# Simulation training in pancreatico-jejunostomy using an inanimate biotissue model improves the technical skills of hepatobiliary-pancreatic surgical fellows

**Ryuji Yoshioka, Hiroshi Imamura, Hirofumi Ichida, Yu Gyoda, Tomoya Mizuno, Yoshihiro Mise, Akio Saiura** *

Department of Hepatobiliary-Pancreatic Surgery, Juntendo University Graduate School, Tokyo, Japan

* a-saiura@juntendo.ac.jp

## Abstract

### Background

Technical proficiency of the operating surgeons is one of the most important factors in the safe performance of pancreaticoduodenectomy. The objective of the present study was to investigate whether surgical simulation of pancreatico-jejunostomy (PJ) using an inanimate biotissue model could improve the technical proficiency of hepato-biliary pancreatic (HBP) surgical fellows.

### Methods

The biotissue drill consisted of sewing biotissues to simulate PJ. The drill was repeated a total of five times by each of the participant surgical fellows. The improvement of the surgical fellows' technical proficiency was evaluated by the supervisor surgeons using the Objective Structured Assessment of Technical Skills (OSATS) scale.

### Results

Eight HBP surgical fellows completed all the 5 drills. Both the OSATS 25 score and OSATS summary score, assessed by the two supervisor surgeons, improved steadily with repeated execution of the PJ drill. The average OSATS score, as assessed by both the supervisor surgeons, improved significantly from the first to the final drill, with a $P$ value of 0.003 and 0.014 for the assessment by the two surgeons, respectively. On the other hand, no chronological alteration was observed in time of procedure ($P = 0.788$).

### Conclusion

Repeated execution of a biotissue PJ drill improved the HBP surgical fellows' technical proficiency, as evaluated by OSATS. The present study lends support to the evidence that simulation training can contribute to shortening of the time required to negotiate the learning curve for the technique of PJ in the actual operating room.

**Data Availability Statement:** All relevant data are within the manuscript and its Supporting Information files.

**Funding:** The authors received no specific funding for this work.

**Competing interests:** The authors have declared that no competing interests exist.

# Introduction

Pancreaticoduodenectomy (PD), usually carried out for malignancies located in the periampullary region or pancreas head, is one of the most complex of abdominal operations, with reported postoperative morbidity and mortality rates of 40% and 0.8%–3%, respectively [1–3]. Strong evidence exists for a volume-outcome relationship, with reports of reduced perioperative morbidity and mortality rates associated with high-risk surgeries, including PD, at high-volume centres [4–8]. Furthermore, some studies have found that the surgeon's volume accounted for a relatively large proportion of the effect of the hospital volume [3, 4]. These results demonstrate, albeit indirectly, that the operating surgeons' technical proficiency plays an important role in the operative outcomes. Birkmeyer et al. provided more direct evidence of the linkage between higher technical skill levels of the surgeons and lower frequency of postoperative complications in bariatric surgery [9], which is a commonly performed, but fairly complex procedure. In relation to PD, it is widely accepted that the chief driver of morbidity following PD is postoperative pancreatic fistula formation (POPF). Recently, Hogg et al. identified the surgeons' technical skill level as an independent predictor of this complication, besides patient-related variables [10].

The finding that more experienced surgeons have superior outcomes suggests the presence of a learning curve for complex technical procedures and surgery. The learning curve needed to attain proficiency in PD has been reported to involve experience of approximately 60 cases [3, 11, 12], although these figures vary according to the procedures. However, it is too difficult for many hepato-biliary pancreatic (HBP) surgeons to negotiate this steep learning curve, because pancreatic resection is a relatively uncommon procedure among various complex surgeries [13, 14]. In fact, a Japanese nationwide administrative database revealed that only 3.7% (31/848) of hospitals performed >28 PDs annually [8]. This result indicates that most HBP surgeons, unfortunately, will perform fewer than half of the requisite number of PJs to successfully negotiate the learning curve during their training. Therefore, it was considered that inanimate simulation training may have an important role if it could be shown to improve the technical proficiency of HBP surgeons that can be applied to actual operations. Tam et al. reported that a curriculum including robotic biotissue reconstructions improved the technical performance of surgical oncology fellows of pancreatico-jejunostomy (PJ) after PD [15]. However, no similar study targeting open PD reconstruction has been conducted yet, to the best of our knowledge.

The Objective Structured Assessment of Technical Skills (OSATS) is widely accepted as a reliable objective method for assessing the surgical skill levels of surgeons, and considered as the gold standard for objective skill assessment [16]. In particular, the global rating scale component of the scale, which consists of several variables scored on a 5-point Likert scale, is widely applicable to surgeons [17]. Accumulation of individual data derived from quantitative estimation of the technical proficiency can easily be used to provide feedback to surgical trainees [18].

Overall, consistently reported relationship between hospital-volume and operative outcome indicated that the care of patients undergoing PD involves variable procedures conducted by a complex team and a role that surgeons' proficiency in the operation room plays may be merely a part of them. In the present study, we specifically focused on the aspect of surgeon's role and investigated the feasibility of simulation training of PJ using an inanimate biotissue model.

# Methods

## Study design

The present study was conducted as an uncontrolled observational prospective study. Two supervisor surgeons (SA and MY) assessed the technical proficiency of eight HBP surgical

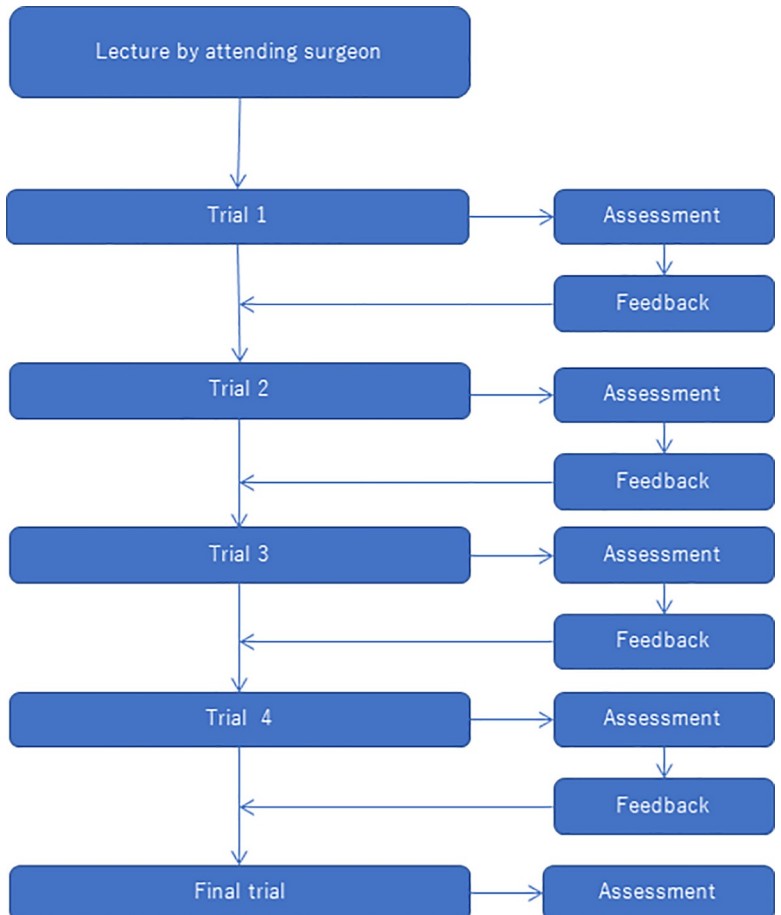

**Fig 1. Flow-chart of the biotissue pancreatico-jejunostomy drill.** In total, 8 HBP SFs repeated the drill 5 times. All the drills were recorded and reviewed, and the performance assessed by two supervisor surgeons. The OSATS score and comments of the supervisor surgeons were provided to the SFs as feedback.

fellows (SFs) in biotissue pancreatico-jejunostomy (PJ) drills. The biotissue PJ drill consisted of the following steps: (1) lecture and demonstration of PJ by one of the supervisor surgeons (SA); (2) drills by the SFs; (3) assessment and review by both supervisor surgeons (SA and MY), (4) supervisor surgeons' feedback to each fellow. This feedback consisted of the OSATS scores and of free comments by the supervisor surgeons without knowledge of who the performer was. Details of assessment by the OSATS scale are provided later in this section (*Outcomes and assessment*). The part of the drill from step (2) to (4) was carried out once per month for a total of five times. The flow chart of this biotissue PJ drill is summarised in Fig 1. The biotissue PJ drills were conducted once a month between September 2019 and January 2020 at the Department of Hepatobiliary-Pancreatic Surgery, Juntendo University Hospital.

## Participants

Inclusion criterion of the present study was as follows: surgical fellows who were affiliated with the Department of Hepatobiliary-Pancreatic Surgery, Juntendo University Hospital as of September 2019. Because all of eight affiliated fellows, thus, candidates, participated in the study from the educational aspect, sample size was not calculated.

### Biotissue PJ model

The inanimate biotissue PJ model used in the present study (Fig 2) was manufactured by FAS-OTEC Co., Ltd., Chiba, Japan. It was composed of polyvinyl alcohol, designed to mimic the pancreas and the jejunum. The internal diameter of the main pancreatic duct was about 1.5 mm.

### Surgical technique

The pancreatic duct and the jejunal mucosal layer were anastomosed in an end-to-side fashion using eight interrupted sutures with single-armed 6–0 polydioxanone. No external drainage tube was inserted. The pancreatic parenchyma and jejunal seromuscular layer were attached by modified Kakita anastomosis, which consisted of four interrupted sutures with double-armed 3–0 polydioxanone. This procedure describes the standard PJ anastomosis technique in PD used at our department, as previously described [19].

### Evaluation of the technical proficiency of the SFs in the PJ drills

Each biotissue drill was recorded and all the videos were reviewed at normal speed by two blinded graders, who were the supervisor surgeons (SA and MY). The technical proficiency in each drill was evaluated in two different ways. First, the technical score was assessed based on the global rating scale component of OSATS, which consists of six variables, as follows: (1) gentleness, (2) time and motion, (3) instrument handling, (4) flow of operation, (5) tissue exposure, and (6) summary score. All the variables were rated on a 5-point Likert scale from 1 to 5 ("1" corresponding to poor and "5" corresponding to excellent). Before the study, the two supervisor surgeons watched several inanimate-model PJ videos together and discussed their scorings for the variables, to standardise the scoring as much as possible. Second, the time for completion of the procedure, defined as the time from the placement of the first stitch to the scission of the last stitch, was determined.

### Outcomes and assessments

The primary endpoint in the present study was the degree of improvement of the technical proficiency in PJ through repeated executions of the biotissue PJ drills. In addition to

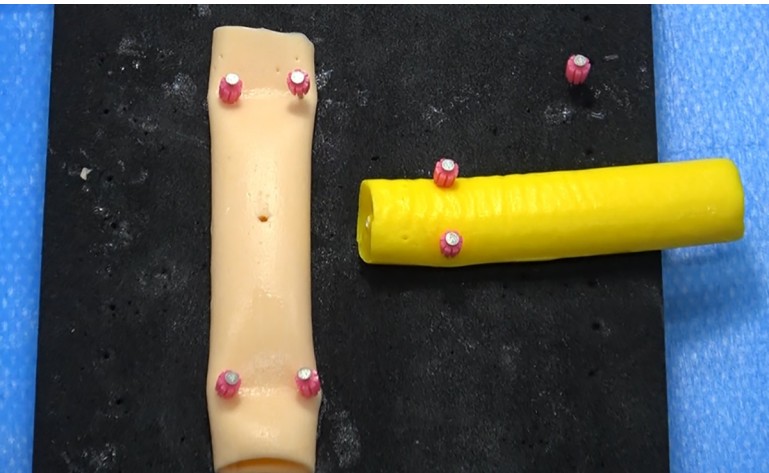

**Fig 2. Pancreatico-jejunostomy (FASOTEC Co., Ltd., Chiba, Japan) model used in the present study.**

comparing the scores for each of the six OSATS variables between the first and fifth attempts, the degree of improvement with repeated executions of the PJ drill was also assessed by OSATS 25, represented by the sum of the scores on the OSATS global rating scale variables 1 to 5 plus the OSATS summary score, i.e., OSATS global rating scale variable 6, and the time taken for the procedure. The variables of the OSATS global rating scale were divided into OSATS 25 and OSATS summary and compared separately according to the report by Hogg et al [10]. The secondary endpoint of this study was to validate the OSATS for assessment of the technical proficiency in an inanimate PJ model, by determining the inter-observer reliability in OSATS scoring between the two graders.

This study was conducted with the approval of the Institutional Ethics Review Board of Juntendo University (#2019015) and is registered in the UMIN-CTR (UMIN000037411, https://www.umin.ac.jp/ctr/index.htm). Written informed consent was obtained from all the participants.

## Statistical analysis

Values are expressed as the means ± SD. The scores for each of the six variables of OSATS were compared between the first and final attempts using the paired t-test. The OSATS 25 score, OSATS summary score, and time taken for completion of the procedure were compared by one-way repeated-measures ANOVA, followed by *post-hoc* test for comparison between each attempt. The inter-observer reliability of the OSATS 25 and OSATS summary assessments between the graders was tested by the interclass correlation test. The significance of the interclass correlation was assessed using average measures with 95% confidence intervals and the *F* test. *P* values of less than 0.05 were considered as being indicative of statistical significance. The statistical analysis was performed using the IBM SPSS software (ver26.0 SPSS Inc., IL, USA).

## Results

### Participants' demographics

The eight HBP SFs were 6 to 17 years from their graduation. The SFs had the experience of performing a median of 6 PDs as operators (0, three SFs; 1, one SF; 11, one SF, 15; one SF; 18, one SF; 19, one SF, respectively) prior to the PJ drills.

### Comparison of the scores on the OSATS variables between the first drill and the last drill

The average scores for the six variables of OSATS in the first drill and final drill as assigned by each grader are summarised in Table 1. The scores for each of the six variables in the final drill were significantly higher than those in the first drill according to the assessments by both graders.

### Alterations of the OSATS 25 score, OSATS summary score, and time for completion of procedure with each repetition of the drill

The mean OSATS 25 score, as assessed by both supervisor surgeons (SA and MY), improved steadily with repeated execution of the PJ drill (Fig 3A and 3B). When the scores in each repeat execution of the drill were compared with those in the first attempt, statistically significant differences from the scores in the first attempt were confirmed for the scores in the last attempt according to the assessment by SA and for the scores in the fourth attempt according to the assessment by MY. Similarly, the OSATS summary scores as assessed by both the supervisor

**Table 1. Comparison of the scores for components of the OSATS between the first and the final attempts.**

| | Grader 1 | | | | | Grader 2 | | | | |
|---|---|---|---|---|---|---|---|---|---|---|
| | First attempt | | Final attempt | | *P* | First attempt | | Final attempt | | *P* |
| | Mean | SD | Mean | SD | | Mean | SD | Mean | SD | |
| Gentleness | 2.4 | 1.1 | 3.6 | 0.7 | 0.002 | 1.9 | 1 | 3.6 | 0.9 | 0.009 |
| Time and Motion | 2.4 | 0.9 | 3.9 | 0.6 | 0.005 | 2.3 | 1 | 3.6 | 0.7 | 0.028 |
| Instrument Handling | 2.3 | 0.9 | 3.6 | 0.7 | <0.001 | 1.8 | 1 | 3.5 | 0.9 | 0.002 |
| Flow of Operation | 2.5 | 0.9 | 4 | 0.5 | 0.001 | 2.3 | 1.2 | 3.4 | 1.1 | 0.019 |
| Tissue Exposure | 2.3 | 1 | 3.6 | 0.5 | 0.008 | 2.1 | 0.8 | 3.8 | 0.9 | 0.003 |
| Summary | 2.3 | 1 | 3.5 | 0.5 | 0.002 | 2.1 | 1 | 3.6 | 0.5 | 0.009 |

OSATS, Objective Structured Assessment of Technical Skills

SD, Standard Deviation

surgeons (SA and MY) also improved with each repeat execution of the PJ drill (Fig 4A and 4B). However, no change in the time taken for completion the procedure was observed with repeated executions of the drill (*P* = 0.788) (Fig 5).

## Inter-observer reliability

Very good inter-observer reliability was noted between the two supervisor surgeons for the OSATS 25 and OSATS summary scoring. The average measures for the OSATS 25 and OSATS summary scores were 0.81 (95% CI: 0.63–0.90, *P* < 0.001) and 0.74 (95% CI: 0.51–0.86, *P* < 0.001), respectively.

## Discussion

The present study showed that repeated execution of the PJ drill in an inanimate biotissue model improved the HBP SFs' technical proficiency, as evaluated by the OSATS score. The OSATS scale is thought to be a feasible and effective tool for the assessment of surgical skills, as judged from the excellent inter-observer reliability for the OSATS 25 and OSATS summary scoring.

The inanimate biotissue PJ drill in the present study consisted of a lecture by one of the supervisor surgeons, followed by repetitions of the PJ drill by the SFs, assessments by the

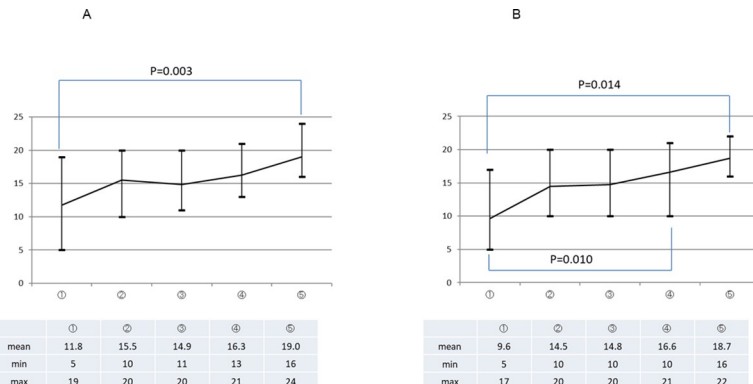

**Fig 3. The average OSATS 25 score as assessed by the two supervisor surgeons.** Y-axis = score out of the maximum of 25; X-axis = attempt number; P values for the differences in the scores between the first and last attempt are listed in addition to those for others with a statistically significant difference.

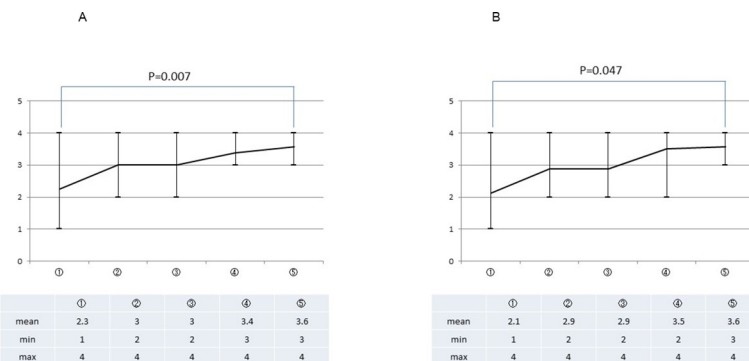

**Fig 4. The average OSATS summary score as assessed by the two supervisor surgeons.** Y-axis = score out of a maximum of 5; X-axis = attempt number; P values for the differences in the scores between the first and last attempt are listed in addition to those for others with a statistically significant difference.

attendings, and feedback. After each drill, the fellows were informed of their own OSATS scores and given advice by the supervisor surgeons. It was thought that these cycles of practice and feedback would aid in improving the SFs' surgical skills. Because POPF formation is the chief driver of morbidity after PD and the surgeon's technical performance is reported as an independent contributory factor to the risk of development of this complication [10], it is quite important for HBP surgeons to acquire proficiency in PJ anastomosis. Because PD is not a commonly performed operative procedure, especially relative to the required number to be performed by each HBP surgeon to negotiate the learning curve [11, 12], the inanimate biotissue drills could help SFs in moving up the learning curve for PJ anastomosis.

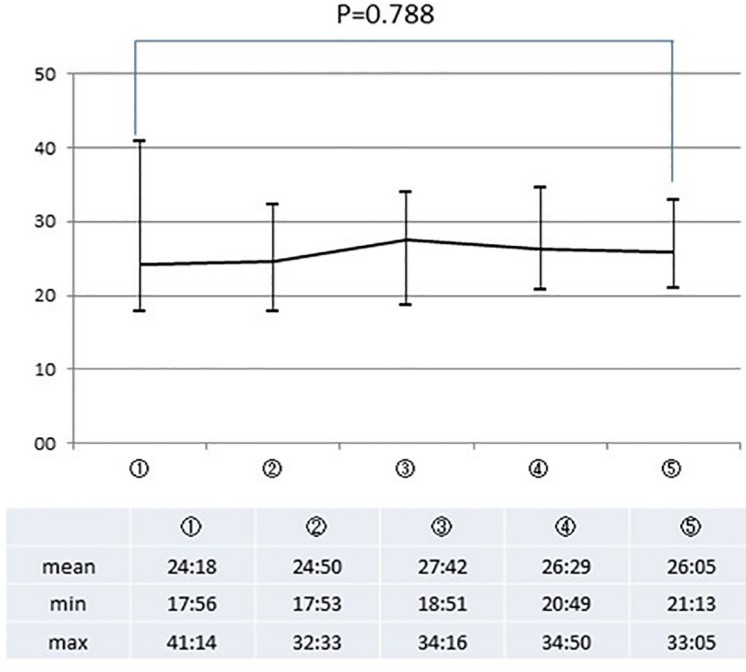

**Fig 5. The average time for completion of the procedure.** Y-axis = time (minute); X-axis = attempt number; the P value for the difference between the first and final attempt is shown. The time listed below the graph is expressed in minutes:seconds.

A few, but excellent, studies have been conducted by the Pittsburg group to investigate the relationship between objectively assessed technical skills and the surgical outcomes in robotic PD [10, 15]. They revealed, for the first time, a direct association between technical proficiency in PJ and the risk of POPF formation. In a subsequent study conducted using an inanimate biotissue model, they demonstrated that technical performance of various reconstructions after robotic PD including PJ, improved through repeated practice. Minimally invasive approaches, especially robotic PD, are fast-growing and promising approaches, however, the reported proportions of procedures performed using a minimally invasive approach and robotic PD are 8.5% and 3.7%, respectively, among 16,222 PDs performed and registered with the American College of Surgeons National Surgical Quality Improvement Program Participant Use Files between 2013 and 2017 [20]. Open PD still remains the mainstay of PD and the present study provides evidence to support the usefulness of including simulation training using a biotissue model in the curriculum for improving a SF's sill in PJ after open PD.

While the various indices of technical skill improved with repeated executions of the PJ drill, the operative time as assessed by Time for procedure, showed no such improvement. Tam and colleagues reported similar findings in their study conducted using an inanimate robotic PJ model [15] and concluded that the time to complete the procedure is the most difficult technical parameter to improve. In support of their contention, Boone et al. demonstrated a moderate learning curve for reducing the operative time than for other metrics in robotic PD [21]. On the other hand, the operative time required for completing PD is reported to be shorter [11] or borderline significantly shorter [3, 12] in high-volume hospitals than in low-volume hospitals. In the present study, the large standard deviation obviously decreased during the drills (Fig 5), suggesting that, at least, the process of standardization of skill in the procedure was under way. Furthermore, both "time and motion" and "flow of operation", which are indices closely related to the "procedure time", improved significantly with repeated execution of the drill. Overall, the number of attempts was possibly insufficient to achieve a statistically significant improvement in the metric of time for completion of the procedure.

The present study had several limitations. First, it remains unclear whether the improvement of the quasi-surgical skills measured by the OSATS in an inanimate biotissue drill could be directly linked to real intraoperative situations. In this regard, Datta et al. reported that the OSATS score assessed using an inanimate model could be extrapolated to actual surgical situations in the operation room [22]. They examined the surgeon's performance in an operation for varicose veins in the leg by the OSATS scale both using an inanimate model and in a real operation and showed a close relationship of the OSATS scores between the two. We also consider that the results of the present study could be extrapolated to real operation situations and help surgeons plan further studies to address this issue directly. Second, the present study adopted only the global rating scale, even though the original OSATS also consists of a task-specific checklist [17] in addition to the global rating scale, because the global rating scale is reported to be superior in terms of the validity and reliability to the task-specific checklist [16, 17]. Although the SFs also received free comments from the two supervisor surgeons in addition to the OSATS score as feedback in the present study, inclusion of procedure-specific assessment using the task-specific checklist may have allowed an even more concrete feedback to be given to the SFs. Finally, this is an observational study and the results need to be interpreted in this context with care. The participants of the present study comprised a small sample size of eight HBP SFs from a tertiary centre who were highly motivated. In addition, there was a large variety in the participants' demographics with respect to years from their graduation (6 to 17 years) and number of PD they had performed (3 to 19 PDs). These could make the results of this study difficult to be interpreted and could have resulted in selection bias,

precluding extrapolation of the results to all types of surgeons in various environments. However, the results of the present study could certainly encourage most surgeons.

In conclusion, repeated execution of a PJ drill in an inanimate biotissue model improved the technical proficiency of the HBP SFs as evaluated using the OSATS scale. The present study lends support to evidence that inclusion of simulation training in the curriculum of SFs can contribute to shortening of the time required for negotiating the learning curve for the PJ technique in the actual operating room.

## Supporting information

**S1 Protocol.**
(DOCX)

**S1 Data.**
(XLSX)

**S2 Data.**
(DOCX)

## Author Contributions

**Conceptualization:** Ryuji Yoshioka, Hirofumi Ichida, Yoshihiro Mise, Akio Saiura.

**Data curation:** Ryuji Yoshioka, Hirofumi Ichida, Yu Gyoda, Tomoya Mizuno.

**Formal analysis:** Ryuji Yoshioka.

**Investigation:** Ryuji Yoshioka, Yu Gyoda, Tomoya Mizuno.

**Methodology:** Ryuji Yoshioka, Hiroshi Imamura.

**Project administration:** Ryuji Yoshioka.

**Resources:** Ryuji Yoshioka.

**Supervision:** Hiroshi Imamura, Yoshihiro Mise, Akio Saiura.

**Writing – original draft:** Ryuji Yoshioka.

**Writing – review & editing:** Ryuji Yoshioka, Hiroshi Imamura, Yoshihiro Mise, Akio Saiura.

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
