## [Decision Letter · Decision Letter 0]

25 Nov 2020

PONE-D-20-31719

Simulation training in pancreatico-jejunostomy using an inanimate biotissue model improves the technical skills of hepatobiliary-pancreatic surgical fellows

PLOS ONE

Dear Dr. Saiura,

Thank you for submitting your manuscript to PLOS ONE. After careful consideration, we feel that it has merit but does not fully meet PLOS ONE’s publication criteria as it currently stands. Therefore, we invite you to submit a revised version of the manuscript that addresses the points raised during the review process.

Please revise according to the issues uncovered by the reviewers.

We look forward to receiving your revised manuscript.

Kind regards,

Academic Editor

PLOS ONE

Journal Requirements:

2) We note that you have stated that you will provide repository information for your data at acceptance. Should your manuscript be accepted for publication, we will hold it until you provide the relevant accession numbers or DOIs necessary to access your data. If you wish to make changes to your Data Availability statement, please describe these changes in your cover letter and we will update your Data Availability statement to reflect the information you provide.

3) Please include captions for your Supporting Information files at the end of your manuscript, and update any in-text citations to match accordingly. Please see our Supporting Information guidelines for more information: http://journals.plos.org/plosone/s/supporting-information.

4)  In your Methods section, please provide additional information about the participant recruitment method and the demographic details of your participants. Please ensure you have provided sufficient details to replicate the analyses such as: a) the recruitment date range (month and year), b) a description of any inclusion/exclusion criteria that were applied to participant recruitment, c) a table of relevant demographic details, d) a description of how participants were recruited, and e) descriptions of where participants were recruited and where the research took place.

5) Please provide a sample size and power calculation in the Methods, or discuss the reasons for not performing one before study initiation.

6) During our internal checks, we noted that your study does not fit the WHO definition of a clinical trial. Therefore, the TREND checklist is not appropriate for this trial design and we suggest removing it from the supplementary data.

Reviewers' comments:

Reviewer's Responses to Questions

**Comments to the Author**

1. Is the manuscript technically sound, and do the data support the conclusions?

Reviewer #1: Yes

Reviewer #2: Yes

2. Has the statistical analysis been performed appropriately and rigorously? 

Reviewer #1: Yes

Reviewer #2: Yes

3. Have the authors made all data underlying the findings in their manuscript fully available?

Reviewer #1: Yes

Reviewer #2: Yes

4. Is the manuscript presented in an intelligible fashion and written in standard English?

Reviewer #1: Yes

Reviewer #2: No

5. Review Comments to the Author

Reviewer #1: It is an excellent manuscript about a very important surgical issue of training. I commend you for writing it in a very elaborate way describing all the requisite details. The results are in line as evident from published research in simulation training. The language is well-written.

Reviewer #2: A very interesting study that supports what has been largely vehiculated in literature, the fact that high-volume centers in the case of PD have better results than small volume centers. It is important to keep in mind and the authors should mention this in the introduction that although the surgeon is one of the determining factors of success but he is not the only one. Caring for a patient with PD usually involves a complex team and although the skills of the surgeon are improved the rest of the team must evolve in the same pace.

A few notes:

It is an observational study, the results need to be interpreted in this context with care.

The authors used the OSATS scale to measure the surgeon evolution. Is there a particular reasons for this scale as there are multiple other options to evaluate surgeon skill.

Line 150-153 since the study did not require patients I do not think this ethics consent is mandatory.

Line 169 the authors mention a large interval of SFs which were 6 to 17 years from their graduation. Also there is a large variety in number of PD each of them performed before this study which varied from 3 PD to 19 PD. Why were the participants included with such a large variety in years of experience and numbers of PD’s. I think this issue needs to be addressed as it can influence the results in a such a small study.

6. PLOS authors have the option to publish the peer review history of their article (what does this mean?). If published, this will include your full peer review and any attached files.

Reviewer #1: **Yes: **Mohammed Amir

Reviewer #2: No

---

## [Author Response · Author response to Decision Letter 0]

12 Dec 2020

Reviewers' comments:

Reviewer's Responses to Questions

Comments to the Author

1. Is the manuscript technically sound, and do the data support the conclusions?

Reviewer #1: Yes

Reviewer #2: Yes

2. Has the statistical analysis been performed appropriately and rigorously?

Reviewer #1: Yes

Reviewer #2: Yes

3. Have the authors made all data underlying the findings in their manuscript fully available?

Reviewer #1: Yes

Reviewer #2: Yes

4. Is the manuscript presented in an intelligible fashion and written in standard English?

Reviewer #1: Yes

Reviewer #2: No

5. Review Comments to the Author

Reviewer #1: It is an excellent manuscript about a very important surgical issue of training. I commend you for writing it in a very elaborate way describing all the requisite details. The results are in line as evident from published research in simulation training. The language is well-written.

Response: We are deeply honored to be highly evaluated from the reviewer 1. 

Reviewer #2: 

Reviewer’s comment: A very interesting study that supports what has been largely vehiculated in literature, the fact that high-volume centers in the case of PD have better results than small volume centers. It is important to keep in mind and the authors should mention this in the introduction that although the surgeon is one of the determining factors of success but he is not the only one. Caring for a patient with PD usually involves a complex team and although the skills of the surgeon are improved the rest of the team must evolve in the same pace.

Response: According the reviewer’s comment, we added the following description in the revised manuscript:

Page 4, line 84-89 (new):

Overall, consistently reported relationship between hospital-volume and operative outcome indicated that the care of patients undergoing PD involves variable procedures conducted by a complex team and a role that surgeons’ proficiency in the operation room plays may be merely a part of them. In the present study, we specifically focused on the aspect of surgeon’s role and investigated the feasibility of simulation training of PJ using an inanimate biotissue model. 

A few notes:

Reviewer’s comment: It is an observational study, the results need to be interpreted in this context with care.

Response: According to the reviewer’s comment, we have revised the manuscript as follows:

Page 13, line 288-289 (new):

Finally, this is an observational study and the results need to be interpreted in this context with care. 

Reviewer’s comment: The authors used the OSATS scale to measure the surgeon evolution. Is there a particular reasons for this scale as there are multiple other options to evaluate surgeon skill.

Response: Although there are various measurement tools for surgical skills as the reviewer commented, a review article by van Hove et al. stated that OSATS is presently accepted as the ‘gold standard’ for objective skill assessment (Br J Surg, 2010). Therefore, we chose OSATS for the measurement of surgeon evolution in the present study. We revised the manuscript of Introduction section as follows:

Page 4, line 77-79 (new):

The Objective Structured Assessment of Technical Skills (OSATS) is widely accepted as a reliable objective method for assessing the surgical skill levels of surgeons, and considered as the gold standard for objective skill assessment19

Reviewer’s comment: Line 150-153 since the study did not require patients I do not think this ethics consent is mandatory.

Response: We appreciate the reviewer’s comment, however, to register this study for the Japanese clinical trial registry as the proof of prospective study, the approval of the institutional review board was mandatory.

Reviewer’s comment: Line 169 the authors mention a large interval of SFs which were 6 to 17 years from their graduation. Also there is a large variety in number of PD each of them performed before this study which varied from 3 PD to 19 PD. Why were the participants included with such a large variety in years of experience and numbers of PD’s. I think this issue needs to be addressed as it can influence the results in a such a small study.

Response: All fellows affiliated with the Department of Hepatobiliary-Pancreatic Surgery, Juntendo University Hospital as of September 2019 participated in the present study. This resulted in a large variety in years of experience and numbers of PDs. According to the reviewer’s comment, we have revised the manuscript as follows:

Page 13, line 291, Page 14, line 292-294 (new):

In addition, there was a large variety in the participants’ demographics with respect to years from their graduation (6 to 17 years) and number of PD they had performed (3 to 19 PDs). These could make the results of this study difficult to be interpreted and could have resulted in selection bias, 

6. PLOS authors have the option to publish the peer review history of their article (what does this mean?). If published, this will include your full peer review and any attached files.

Do you want your identity to be public for this peer review? For information about this choice, including consent withdrawal, please see our Privacy Policy.

Reviewer #1: Yes: Mohammed Amir

Reviewer #2: No

---

## [Decision Letter · Decision Letter 1]

21 Dec 2020

Simulation training in pancreatico-jejunostomy using an inanimate biotissue model improves the technical skills of hepatobiliary-pancreatic surgical fellows

PONE-D-20-31719R1

Dear Dr. Saiura,

We’re pleased to inform you that your manuscript has been judged scientifically suitable for publication and will be formally accepted for publication once it meets all outstanding technical requirements.

Kind regards,

Academic Editor

PLOS ONE

Additional Editor Comments (optional):

Reviewers' comments:

Reviewer's Responses to Questions

**Comments to the Author**

1. If the authors have adequately addressed your comments raised in a previous round of review and you feel that this manuscript is now acceptable for publication, you may indicate that here to bypass the “Comments to the Author” section, enter your conflict of interest statement in the “Confidential to Editor” section, and submit your "Accept" recommendation.

Reviewer #1: All comments have been addressed

Reviewer #2: All comments have been addressed

2. Is the manuscript technically sound, and do the data support the conclusions?

Reviewer #1: Yes

Reviewer #2: Yes

3. Has the statistical analysis been performed appropriately and rigorously? 

Reviewer #1: Yes

Reviewer #2: Yes

4. Have the authors made all data underlying the findings in their manuscript fully available?

Reviewer #1: Yes

Reviewer #2: Yes

5. Is the manuscript presented in an intelligible fashion and written in standard English?

Reviewer #1: Yes

Reviewer #2: (No Response)

6. Review Comments to the Author

Reviewer #1: As submitted earlier I have nothing to add but you have addressed the concerns raised by other worthy reviewer.

Reviewer #2: The authors addressed the raised issues with the article and from my personal point of view it is fit for publication.

7. PLOS authors have the option to publish the peer review history of their article (what does this mean?). If published, this will include your full peer review and any attached files.

Reviewer #1: No

Reviewer #2: No

---

## [Editor Report · Acceptance letter]

5 Jan 2021

PONE-D-20-31719R1 

Simulation training in pancreatico-jejunostomy using an inanimate biotissue model improves the technical skills of hepatobiliary-pancreatic surgical fellows 

Dear Dr. Saiura:

I'm pleased to inform you that your manuscript has been deemed suitable for publication in PLOS ONE. Congratulations! Your manuscript is now with our production department. 

Kind regards, 

on behalf of

Dr. Robert Jeenchen Chen 

Academic Editor

PLOS ONE